# Home-Based Measurements of Dystonia in Cerebral Palsy Using Smartphone-Coupled Inertial Sensor Technology and Machine Learning: A Proof-of-Concept Study

**DOI:** 10.3390/s22124386

**Published:** 2022-06-09

**Authors:** Dylan den Hartog, Marjolein M. van der Krogt, Sven van der Burg, Ignazio Aleo, Johannes Gijsbers, Laura A. Bonouvrié, Jaap Harlaar, Annemieke I. Buizer, Helga Haberfehlner

**Affiliations:** 1Rehabilitation Medicine, Amsterdam UMC Location Vrije Universiteit Amsterdam, 1081 HZ Amsterdam, The Netherlands; dylan.den.hartog@hotmail.com (D.d.H.); m.vanderkrogt@amsterdamumc.nl (M.M.v.d.K.); l.bonouvrie@amsterdamumc.nl (L.A.B.); ai.buizer@amsterdamumc.nl (A.I.B.); 2Amsterdam Movement Sciences, Rehabilitation and Development, 1081 BT Amsterdam, The Netherlands; 3Netherlands eScience Center, 1098 XH Amsterdam, The Netherlands; s.vanderburg@esciencecenter.nl; 4Moveshelf Labs B.V., 3521 AL Utrecht, The Netherlands; ignazio.aleo@moveshelf.com (I.A.); johannes.gijsbers@moveshelf.com (J.G.); 5Department Biomechanical Engineering, TU Delft, 2628 CD Delft, The Netherlands; j.harlaar@tudelft.nl; 6Emma Children’s Hospital, Amsterdam UMC Location University of Amsterdam, 1105 AZ Amsterdam, The Netherlands; 7Department of Rehabilitation Sciences, KU Leuven, Campus Bruges, 8200 Bruges, Belgium

**Keywords:** cerebral palsy, dystonia, choreoathetosis, machine learning, home-based, inertial measurement unit, wearable device

## Abstract

Accurate and reliable measurement of the severity of dystonia is essential for the indication, evaluation, monitoring and fine-tuning of treatments. Assessment of dystonia in children and adolescents with dyskinetic cerebral palsy (CP) is now commonly performed by visual evaluation either directly in the doctor’s office or from video recordings using standardized scales. Both methods lack objectivity and require much time and effort of clinical experts. Only a snapshot of the severity of dyskinetic movements (i.e., choreoathetosis and dystonia) is captured, and they are known to fluctuate over time and can increase with fatigue, pain, stress or emotions, which likely happens in a clinical environment. The goal of this study was to investigate whether it is feasible to use home-based measurements to assess and evaluate the severity of dystonia using smartphone-coupled inertial sensors and machine learning. Video and sensor data during both active and rest situations from 12 patients were collected outside a clinical setting. Three clinicians analyzed the videos and clinically scored the dystonia of the extremities on a 0–4 scale, following the definition of amplitude of the Dyskinesia Impairment Scale. The clinical scores and the sensor data were coupled to train different machine learning models using cross-validation. The average F1 scores (0.67 ± 0.19 for lower extremities and 0.68 ± 0.14 for upper extremities) in independent test datasets indicate that it is possible to detected dystonia automatically using individually trained models. The predictions could complement standard dyskinetic CP measures by providing frequent, objective, real-world assessments that could enhance clinical care. A generalized model, trained with data from other subjects, shows lower F1 scores (0.45 for lower extremities and 0.34 for upper extremities), likely due to a lack of training data and dissimilarities between subjects. However, the generalized model is reasonably able to distinguish between high and lower scores. Future research should focus on gathering more high-quality data and study how the models perform over the whole day.

## 1. Introduction

Cerebral palsy (CP) is the most common physically disabling condition in childhood, associated with a lifelong movement disability [1]. CP is caused by brain malformation or brain injury acquired pre- or perinatally, or early in the postnatal period [1]. Three different subtypes of motor disorder are distinguished within CP: spastic, dyskinetic and ataxic CP [1]. Dyskinetic CP accounts for 6–15%, with a prevalence of about 0.12–0.3 in every 1000 live births in Europe [2]. On average, dyskinetic CP is the most disabling form of CP [3]. 

Dyskinetic movements and postures are characterized by two features, which often co-exist in the same patient: dystonia, which is described by abnormal patterns of posture and/or slow movements; and choreoathetosis, which is characterized by faster involuntary, uncontrolled, jerky, and contorting movements [3]. Dystonia and choreoathetosis can seriously hamper everyday activities of patients. 

Several new invasive treatments have been developed in recent decades to reduce dyskinesia and thereby improve daily live function [3]. These neuromodulation treatments include intrathecal baclofen treatment provided via an implanted microinfusion pump, and deep brain stimulation [4]. Intrathecal baclofen treatment has been shown to effective in achieving personal treatment goals in children with dyskinetic CP, however only a small effect on dystonia and choreoathetosis could be shown [5,6]. Deep brain stimulation can be effective in a selected group of individuals with dyskinetic CP. For patient selection, an in-depth understanding of dyskinetic movements is required [7].

Accurate and reliable measurements of the severity of the movement disorder are essential for indication, evaluation, monitoring and fine-tuning of these treatments (i.e., indication, dosage of medication and location and dosage of stimulation). However, it remains a huge challenge to capture the severity of the dyskinetic movements and postures in an objective way.

Assessment of dystonia and choreoathetosis in children and adolescents is now commonly performed by visual evaluation either directly in the doctor’s office or from video recordings using standardized scales [8]. These assessments are both lacking objectivity and require much time and effort of clinical experts. Furthermore, only a snapshot of the severity of dyskinetic movements is captured, and they are known to fluctuate over time and can increase with fatigue, pain or emotions (e.g., stress), which likely happens in a clinical environment [9]. 

The gold standard for the analysis of upper and lower extremity movements in individuals with CP is a collection of 3D kinematics (rotations of multiple joints and segments during reaching, grasping and walking) [10,11]. However, the collection of data for 3D kinematics requires advanced motion capture systems, which do not allow outside-lab measurements. To enable measurements at home and in daily life environments, for CP, there is an increasing interest in using simpler systems for data collection, such as video-based markerless motion tracking (e.g., OpenPose) [12] and Inertial Measurements Units (IMUs) [13,14]. These easily applicable measurement systems, combined with machine learning models trained by algorithms (e.g., traditional such as logistic regression, random forest, support vector machine, or deep learning algorithms) may significantly contribute to the early detection of CP [15] and the monitoring of daily life functions [12,13]. Within other neurological diseases such as Parkinson’s disease [16,17,18] and Huntington’s disease [19], wearable sensors in combination with machine learning techniques are also increasingly used in monitoring of movement disorders. Specifically, in the last decade, IMUs became an attractive and accurate solution, with an increased battery life of several hours, or days, small form-factors, and low cost, making them a very suitable option for home-based measurements for the assessment of movement disorders in childhood. Inertial motion quantities, such as accelerations and angular velocities, combined with an algorithm that automatically assess the presence and amplitude of dystonia and choreoathetosis, would yield meaningful information without manual and time investment. However, no algorithm specific for automatic evaluation of dystonia and choreoathetosis from sensor data is available for dyskinetic CP. As dystonia and choreoathetosis can be significantly variable in dyskinetic CP between as well as within subjects concerning involved body parts, and dependent on environmental factors and the activity performed [20,21,22], the automatic evaluation is a challenging machine learning task.

Monitoring of movement disorders of children and young adults with dyskinetic CP for a longer period of time within a well-known environment would provide a realistic and reliable evaluation of dystonia and choreoathetosis and can serve treatment decision and monitoring for this complex group. Within this proof-of-concept study, we used four IMUs coupled to a smartphone, allowing the collection of IMUs data and time-synchronized video recordings at home. We aim (1) to show the feasibility of data collection in a natural environment in children and young adults with dyskinetic CP and (2) to train a machine learning model that can detect and score dystonia using IMU data.

## 2. Materials and Methods

The flowchart in Figure 1 summarizes the dataflow from home measurements (IMUs and videos) towards the final evaluation of the picked classification models. Below, a detailed description of the methods is provided.

### 2.1. Participants

Participants were recruited from the pediatric outpatient rehabilitation department during regular appointments from 1 March till 31 October 2021. Patients were included if they had: (1) a clinical diagnosis of dyskinetic CP [23,24], (2) were 4–24 years old, and (3) if parents/caregivers were able to follow the instructions for the home-based measurements.

In total, 12 participants were included; Participants had following characteristics (mean ± Standard deviation (range)): Age 14.9 ± 4.4 (10.2–21.4) years;Weight: 37.3 ± 17.2 (21.7–76.9) kg;Height: 145.41 ± 23.5 (116–190) cm;4 females/8 males;Gross Motor Function Classification System (GMFCS): II (*n*= 2), IV (*n* = 5) or V (*n* = 5);Manual Ability Classification System (MACS): II (*n* = 1), III (*n* = 3), IV (*n* = 2), V (*n* = 6).

The study was approved by the Medical Ethics Committee of the VU University Medical Center Amsterdam (The Netherlands). Written informed consent was obtained from participants and, if applicable, their parents for participation in this study.

### 2.2. Measurements

#### 2.2.1. Materials

The following materials were used for the experiments:(1)Mobile phone: Samsung A71 (Samsung Electronics, Daegu, South-Korea), with;(2)MODYS@home app (developed by Rutgers Engineering, Norg, The Netherlands): a custom mobile application for Android, using the Xsens DOT Software Development Kit (SDK). The app automatically links recorded videos with corresponding time stamps in the sensor data;(3)Four IMUs (Xsens DOT, Xsens Technologies B.V., Enschede, The Netherlands). Xsens DOT is a wearable sensor incorporating 3D accelerometers, gyroscopes and magnetometers to provide acceleration, angular velocity, and the Earth’s magnetic field. Combined with Xsens, sensor fusion algorithms, 3D orientation and free acceleration are provided [10]. Inertial and orientation data outputs of the Xsens DOT sensor are presented in Table 1. The Xsens DOT sensors were set to measure with a sampling frequency of 60 Hz with an accelerometer range of ±16 g and a gyroscope range of ±2000 dps;(4)Fixation material (Xsens DOT Adhesive patches (Xsens DOT, Xsens Technologies B.V., Enschede, The Netherlands), FabriFoam NuStim Wrap (Fabrifoam, Exton, PA, USA), 3m Coban self-adherent wrap (3M, St. Paul, MN, USA).

#### 2.2.2. Procedure

For the measurements within this proof-of-concept study, participants could choose between measurements at home or in the hospital. For home measurements, participants received a measurement set containing a mobile phone with the MODYS@home app installed, four IMUs, chargers for the phone and sensors, fixation material and a manual. The four Xsens DOT sensors were attached towards the forearm (palmar on the forearm, proximal of processus styloideus ulnae) and lower leg (proximal of the lateral malleolus) (Figure 2). The method of fixation on the attachment site was individually determined. Participants and parents/caregivers were instructed on how to place the IMUs on the participant and how to use the MODYS@home app to record videos and collect sensor data. They were asked to record 10 videos of about 1 minute each day, for both active and resting situations for 7 days within a period of 2 weeks. After the period of 2 weeks, the measurements set was picked up by the researcher and data was transferred by USB-connection for further analysis. For the individuals measured in the hospital, activities and rest data were collected, mimicking a home-based environment. Examples of activities performed at home as well as in the hospital are wheelchair driving, walking, stair climbing, cycling, eating/drinking, sport activities, gaming, computer use, playing music, playing a board game, reading, watching a video/television, using a communication device and resting in a chair or lying down. Activities were chosen by parents/caregivers and participants dependent on the functional level of the individual. Videos during passive movements, e.g., caregiving, transfers were excluded in the current analysis.

### 2.3. Software

Clinical scoring was done using an open-source tool for video annotation, ELAN version 6.2, Max Planck Institute for Psycholinguistics, Nijmegen, The Netherlands, sourced from: https://archive.mpi.nl/tla/elan/download (accessed on 3 May 2022); MATLAB (Mathworks Inc., Natick, MA, USA) release R2018b was used for processing the data and developing the machine learning models. The code used in the current study is made available (Appendix A).

### 2.4. Clinical Scoring

Three clinicians assessed the videos. For each time window of 5 seconds, a score between 0–4 was assigned for dystonia for the left and right arm and the left and right leg, separately, following the definition of amplitude of the Dyskinesia Impairment Scale (DIS) [25] for scoring dystonia. Within Parkinson’s disease, a 5 s time windows was found to be optimal to achieve the minimum estimation error when estimating the severity of tremor, bradykinesia and dyskinesia using accelerometers and machine learning [26]. The DIS distinguishes between proximal and distal segments of the extremities when scoring amplitude. This score was summarized within the current scoring. Thus, each clinician provided four scores for dystonia for each time window of each video. The median of the three scores was calculated as the final score for the machine learning.

### 2.5. Data Pre-Processing 

Data from the IMUs required pre-processing to serve as input for machine learning. As some time stamps were missing with different sensors, the sensor data from the four sensors was synchronized using linear interpolation with the values from adjacent timestamps.

For each sensor, the resultant free acceleration (a) and resultant angular velocity (ω) at each time stamp was calculated using Equations (1) and (2) respectively:(1)$$a_{r}=\sqrt{a_{x}^{2}+a_{y}^{2}+a_{z}^{2}}$$
(2)$$\omega_{r}=\sqrt{\omega_{x}^{2}+\omega_{y}^{2}+\omega_{z}^{2}}$$

Each sensor therefore provided 11 signals: 4 accelerations, 4 angular velocities and 3 Euler angles. A single timestamp containing data from all four sensors consisted of 4 × 11 = 44 signals. Each 5 s time window contained 300 timestamps.

In MATLAB, the videos were automatically linked to the sensor data, cutting out parts of the sensor data where a video was recorded. These cut-out parts of sensor data were segmented into time windows of 5 s, equal to the clinical windows. Finally, the clinical scores were automatically linked to the corresponding time windows. Figure 3 shows an example of the sensor signals togethers with the clinical scoring.

Per subject, two tables containing input data and output were created for machine learning. Tables were created for both upper and lower extremities, by adding the data from the left and right extremities. 

### 2.6. Feature Selection and Extraction

Research has shown that feature selection is an effective way to improve the learning process and recognition accuracy, and decreases the complexity and computational cost [27]. We used a method recently described by Den Hartog et al. [28]. In brief, time domain and frequency domain features were tested on the data from all subjects. A Fast Fourier Transform was used to extract frequency-domain features. Initially, 32 different feature classes were tested for usability. For each time window, a single feature class was extracted per IMU signal, creating 11-dimensional feature vectors (1 feature class × 11 signals). These feature vectors were then fed to six different machine learning algorithms (Decision Tree, Discriminant Analysis, Naïve Bayes, Support Vector Machine, k-nearest neighbors, and Ensemble Learning), to test the feature classes’ predictive power. Feature classes were only selected if they were capable of achieving an F1 score of at least 0.7 with a machine learning algorithm, indicating a strong correlation with the output. A total of 10 feature classes passed the selection round (Table 2).

Next, for each time window, all 10 feature classes were extracted for each of the 11 IMU signals, creating 10-dimensional feature vectors (10 classes × 11 signals). This means that for each time window there are 110 features that could describe the characteristics of that window.

Next, sequential feature selection (SFS) as described by MATLAB (Sequential Feature Selection—MATLAB & Simulink—MathWorks Benelux) was used, as this is an effective way to identify redundant and irrelevant features. Sequential feature selection is a wrapper-type feature selection algorithm that starts training using a subset of features and then adds or removes a feature using a selection criterion. The selection criterion directly measures the change in model performance that results from adding or removing a feature. The algorithm repeats training and improving a model until its stopping criteria are satisfied.

In this study, sequential feature selection (SFS), with a maximum number of objective evaluation of 20, was used. SFS sequentially adds features to an empty candidate set until the addition of further features does not decrease the objective function. In this study, misclassification rate was set as the objective function. 

Finally, the extracted features were normalized to rescale the data to a common scale. Supervised machine learning algorithms learn the relationship between input and output and the unit, scale, and distribution of the input data may vary from feature to feature. This will impact the classification accuracy of the models. In this work, the data was normalized by scaling each input variable to a range of 0 to 1.

### 2.7. Machine Learning and Algorithms

After processing the data and extracting features, the next step is to feed the feature vectors to machine learning algorithms. In this study, six types of supervised machine learning algorithms were tested: Decision Tree, Discriminant Analysis, Naïve Bayes, Support Vector Machine, k-nearest neighbors, and Ensemble Learning.

### 2.8. Training, Validating and Testing

For an objective evaluation of the machine learning algorithms, the datasets were divided into a training dataset, validation dataset and testing dataset.

Since the datasets were small, a 5-fold cross-validation was used to evaluate the performance of the models. For each iteration, 80% of the data was used for training and validation, and 20% was used for testing. For training the machine learning models, another 5-fold cross-validation was also used within the training and validation data.

The validation dataset provides an evaluation of a model fit on the training dataset while tuning the model’s hyperparameters [29]. After training and validating, the trained models were evaluated with the testing data containing 20% of the data. The testing dataset was used to provide an unbiased evaluation of a final model fit on the training dataset [29]. This testing dataset was not used for training. Since a 5-fold cross-validation was used, all samples were tested in the testing dataset. The models’ predicted clinical scores of the testing data were compared with the true clinical scores, to calculate the precision, recall and F1 score of the model when used on unseen data [29].

Most datasets contain a severe skew in the class distribution, which could lead to the machine learning algorithms performing poorly on the minority classes. To address this problem, the training data was oversampled to equalize the number of samples per score. 

Different models were trained, validated, and tested using four different settings for each type of the six machine learning algorithms. This was done for both the upper extremities dataset and the lower extremities dataset. Models were trained (1) using all features (ALL), (2) using all features and hyperparameter tuning to find the optimal set of hyperparameters (ALL + HYP), (3) with selected features (SFS) and (4) using selected features and hyperparameter tuning (SFS + HYP) (Table 3).

For the ALL + HYP and SFS + HYP, the hyperparameters were determined using a Bayesian optimization algorithm with 15 iterations during the first fold (Table 3). The found hyperparameters were then used during the remaining folds to test for the model’s precision, recall and F1 score. 

Individual models (i.e., using the data of one participant only) as well as generalized models (i.e., using all data) were trained. The performance for each model was calculated. The trained individual models were tested on holdout testing data using 5-fold cross-validation. Generalized models were evaluated using leave-two-subjects-out cross-validation (6-fold). For each of the 6 folds, data from 10 subjects was used for training and validating (5-fold cross-validation), and tested on the data from the two left-out subjects. 

As main performance metric the F1 score was computed, which used the precision and recall (Equations (3)–(5)), calculated from ‘True positive’ (TP), ‘False positive’ (FP), and ‘False negative’ (FN) scores. F1 scores were calculated after training and validating, and after testing the models on the holdout test data. Per patient, the models with the highest F1 scores were selected as the final models for that patient. In addition, for the generalized models the root mean square errors (RMSE) was calculated and confusion matrix plotted for better interpretation of the model performance.
(3)$$\mathrm{precision}=\frac{\mathrm{TP}}{\mathrm{TP}+\mathrm{FP}}$$
(4)$$\mathrm{recall}=\frac{\mathrm{TP}}{\mathrm{TP}+\mathrm{FN}}$$
(5)$$F1\;\mathrm{score}=2\cdot \frac{\mathrm{precision}\cdot \mathrm{recall}\;}{\mathrm{precision}+\mathrm{recall}}$$

## 3. Results

### 3.1. Datasets

Two patients were measured within the movement laboratory mimicking a home environment and activities, the other ten patients were measured at home by parents/caregivers. Even though parents/caregivers were instructed to record 10 one-minute videos each day, there were large differences in the number of samples (5-s time windows) in the final datasets for each subject. Not all parents/caregivers filmed as many videos as they were instructed. One participant stopped after one measurement due to uncomfortableness while attaching and wearing the sensors. The data of this subject were excluded for the individual trained models. Furthermore, errors in the sensors occurred for some measurements, resulting in loss of data. The most common errors were failure of one or more sensors and an error in the synchronization between the sensors. Moreover, not all windows could be scored because certain body parts were not visible on the videos. These factors led to different sizes of datasets for each subject. Table 4 lists the number of samples in each dataset of each subject. See Appendix A for an overview of the distribution of the scores for each patient. The full dataset is available (Appendix A).

### 3.2. Individual Clinical Scores Classification

Table 4 gives an overview of the best models (algorithm and model type) of each patient, together with the corresponding F1 scores, precision and recall. k-nearest neighbors algorithms led to the highest F1 validation score in most datasets and were therefore most often chosen as final model. Table 5 gives and overview of the mean F1 scores, precision and recall of all best models combined. High F1 scores (0.97 ± 0.03 for lower extremity dystonia and 0.93 ± 0.06 for upper extremity dystonia) were observed during validation of the individual models. In the independent test datasets, the F1 scores (0.67 ± 0.19 for lower extremity dystonia and 0.68 ± 0.14 for upper extremity dystonia) were lower (Table 5). 

### 3.3. Generalized Clinical Scores Classification

See Table 6 for an overview of the best models per dataset. Figure 4 and Figure 5 show the confusion matrices of the datasets. The generalized model showed lower F1 scores (0.45 for the lower extremities and 0.34 for the upper extremities) in the test datasets than the individual models. F1 scores were high in the validation data sets, but significantly lower in the test data sets, indicating the model does not work equally as well on unseen data. The majority of misclassifications occurred in neighbouring clinical scores, since they present similar behaviours. The RMSE were 1.07 for dystonia lower extremties and 0.98 for dystonia upper extremites, respectively. A clinical score of 4 in the dystonia upper extremities data set was never correcly classified, likely due to a lack of training samples during training of the models.

## 4. Discussion

Within this study, the feasibility was assessed to train machine learning models with a sufficient performance within dyskinetic CP by using home-based measured IMU and video data, collected by parents/caregivers. 

In summary, most of the parents/caregivers were able to collect enough data to clinically score the videos and use IMUs data for feature calculation. For 1 patient out of 12, discomfort due to the fixation of sensors was reported. We consider the performance (i.e., F1 score) of the individual trained model as moderate and the overall performance of the generalized models as low. However, when looking at the confusion matrices, the misclassifications were most often observed in neighboring classes, indicating that these models are reasonably able to correlate between the severity of the disorder and the clinical score. This observation is confirmed by the RMSEs of about 1 on a 4-point scale. 

The current results are in line with previous studies using wearable IMUs or accelerometers within other patient populations (e.g., Parkinson’s disease [30,31,32] and Huntington’s disease [19], showing that it is feasible to automatically predict the severity of movement disorders such as tremor, bradykinesia and dyskinesia. Most studies using wearables to monitor movement disorders have been performed within Parkinson’s disease including steps towards clinical implementation (i.e., assessment of measurement properties of methods). However, widespread clinical use is still lacking [16,18]. When relating the current results to studies in Parkinson’s disease, reported performance are comparable: e.g., Tsipouras et al. [31] used IMUs to automatically classify lepodova-induced dyskinesia within standardized tasks on a 0–4 scale, using machine learning algorithms and multiple combinations of sensors and features. A generalized model within this study achieved an average accuracy (79% ± 11%) [31]. However, the results need to be interpreted with care as no independent test set was used and no F1 scores were computed. Another study used sensors placed on the upper and lower extremities. A high correlation between the estimated dyskinesia severity scores was found between the model prediction and the expert-rated scores on (r = 0.77 (*p* < 0.001)) [33]. Although the population of Parkinson’s disease and dyskinetic CP are not directly comparable, it indicates the potential of the proposed methodology for individuals with dyskinetic CP. A current study suggested that IMUs can be used as a mobile alternative for marker-based motion capture (omitting the need for an advanced movement laboratory) within upper extremity movements analysis of standardize movements in dyskinetic CP [14]. The proposed methodology goes one step further by using home-based collected IMUs data within unstandardized situations. This methodology is especially interesting for individuals who cannot perform standardized movements (such as gait and reaching/grasping), where instrumented methods are lacking [34]. In addition, the methodology gives the opportunity to capture the variability within dystonia for a longer period of time. The results show that within an individual, dystonia is ‘consistent’ enough to be detected within unseen data. However, this is not true for all individuals (i.e., subject 11 and subject 12 showed lower F1 scores within the test set). The same applies to the generalized models. A possible inconsistency within data of individuals as well as between individuals could be explained by, on the one hand, the challenge for clinicians to score home-based videos consistently and, on the other hand, the variation of dystonia concerning velocity and position that can occur within and between individuals [20]. As the performance of machine learning models greatly depends on the amount, the coverage and quality of the data, the performance of individual model would most likely increase with the collection of more data from each individual, as well as measures from more patients. 

A limitation of this study is the low number of subjects included, which limits the amount of data used to train and test the generalized models. Another limitation of this study is that data was collected only at certain fixed moments, which were mainly standing, sitting, and lying down. The developed models are therefore not properly trained with data from other everyday activities. This is likely to lead to inaccuracies in the predictions if the models are used to predict data over an entire day. Future research should focus on gathering more types of movements and activities, to train even more accurate predictive models. The model might also improve by adding IMUs data from children and young adults without a movement disorder, especially as it has been hypothesized that overflow movements seen in dystonia may contain a small repertoire of involuntary movements within a more variable repertoire of intended voluntary movements [35]. As collection of more and variable data might be difficult to perform on large scale, possible data augmentation techniques for time series should be considered in future studies [36]. In addition, it could be an option to perform a ‘calibration measurement’ for each individual before using sensors in a home environment [16], add some extra clinical scores on time windows and use transfer learning (i.e., adding the individual scored data to the pre-trained generalized model) to improve the performance of the generalized models for each patient individually. 

Since the results of this study demonstrated the feasibility of monitoring dystonia at home, it would be interesting to study the use of the models for treatment assessment (e.g., how the clinical scores vary before and after intrathecal baclofen treatment), with the hypothesis that the clinical scores will decline after the treatment. Moreover, the methods described in this paper could also be used to classify choreoathetosis, which also occurs in dyskinetic CP. However, there was too little variation in the scores in the current data to train models to classify choreoathetosis.

## 5. Conclusions

The results of this study indicate that it is feasible to assess dystonia in dyskinetic CP outside a clinical setting, using home measurements and individually trained machine learning models and thereby provide clinical useful information about the progression of dystonia during a longer period of time. The findings are in line with previous research on automatic assessment of dyskinesia in Parkinson’s disease. To enhance clinical care, future studies should evaluate how standard dyskinetic CP measures can be complemented by providing frequent, objective, real-world assessments. Even though the generalized models achieved low F1 scores, they are reasonably able to link high clinical scores to high severity of the disorder and vice versa, even though they were trained with a limited amount of data. Future research should focus on gathering more high-quality data and study how the models perform over longer periods of time. 

## Figures and Tables

**Figure 1 sensors-22-04386-f001:**
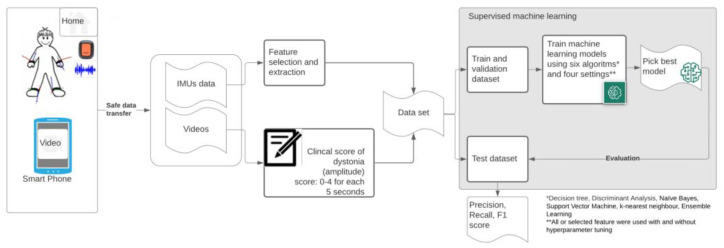
Flowchart of used methodology—measurements of dyskinesia at home (MODYS@home).

**Figure 2 sensors-22-04386-f002:**
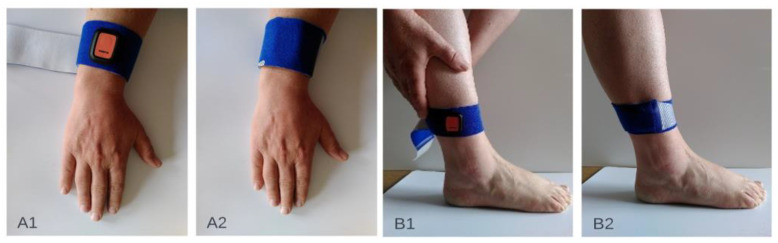
Attachment sites of the inertial sensor units on the upper extremity (**A1**,**A2**) and the lower extremity (**B1**,**B2**).

**Figure 3 sensors-22-04386-f003:**
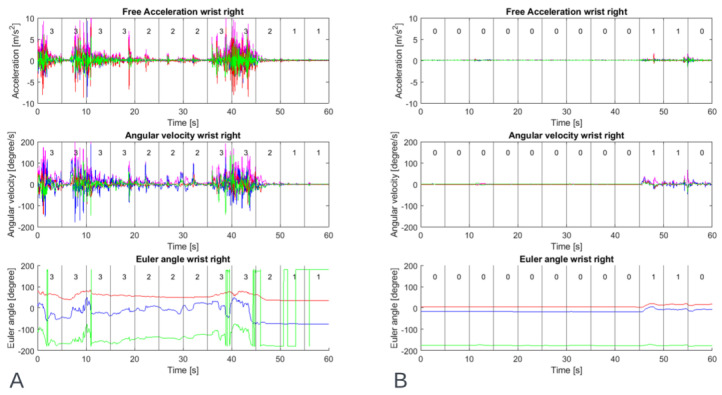
Example of data of one participant’s right wrist during a resting activity showing inertial sensor output: free acceleration, angular velocity and Euler angle, (**A**) with a high level of dystonia and (**B**) with a low level of dystonia. The number within the time windows of 5 s is the median clinical score of three raters for upper extremity dystonia of the right wrist.

**Figure 4 sensors-22-04386-f004:**
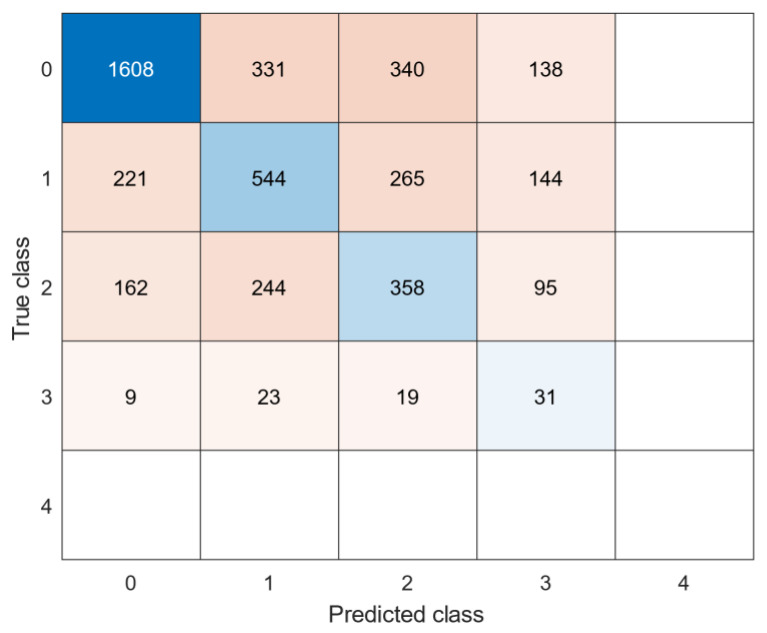
Confusion matrix: generalized model of lower extremities dystonia.

**Figure 5 sensors-22-04386-f005:**
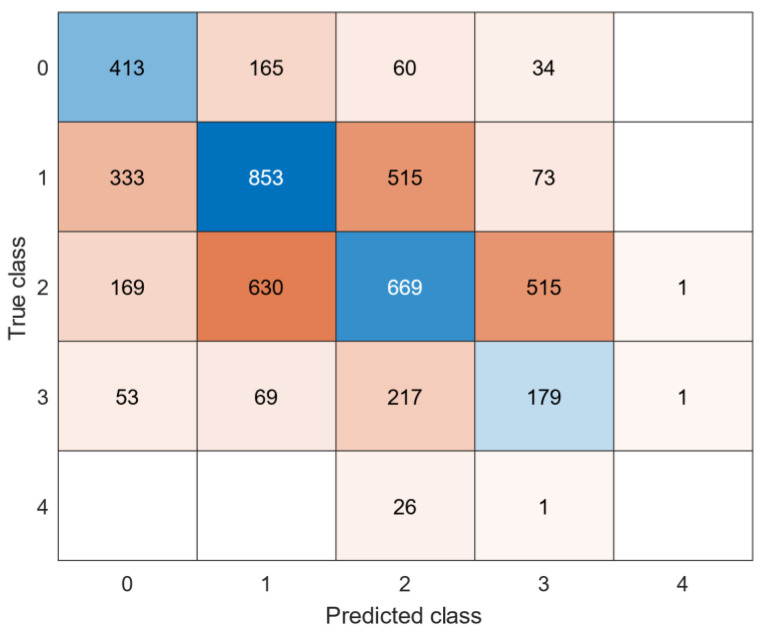
Confusion matrix: generalized model of upper extremities dystonia.

**Table 1 sensors-22-04386-t001:** Inertial and orientation data outputs of Xsens DOT sensors.

Output	Unit
Free acceleration	m/s^2^
Angular velocity	degree/s
Euler angles	degree. Roll, pitch, yaw (ZYX Euler Sequence. Earth fixed type, also known as Cardan or aerospace sequence)

**Table 2 sensors-22-04386-t002:** Overview of the 10 feature classes passed the feature selection round.

Nr	Feature Class
1	Absolute harmonic mean
2	Absolute maximum
3	Bandpower
4	Geometric mean
5	Maximum
6	Median
7	Minimum
8	Root-mean-square
9	Root-sum-of-squares
10	Shannon entropy

**Table 3 sensors-22-04386-t003:** Types of models used for training, validating, and testing.

Model	Features	Hyperparameter Tuning
ML model (ALL)	All features	no
ML model (ALL + HYP)	All features	yes
ML model (SFS)	Selected features with SFS	no
ML model (SFS + HYP)	Selected features with SFS	yes

ML = machine learning; ALL = all features; SFS = Sequential feature selection; HYP = hyperparameter tuning.

**Table 4 sensors-22-04386-t004:** Overview of best individual model per dataset for each patient.

Subject	Dataset	Samples	Best Algorithm	Model	F1 Score Validation	F1 Score Test	Precision Test	Recall Test
Subject 1	dys lower	720	KNN	ALL + HYP	1	0.50	0.98	0.33
	dys upper	726	KNN	SFS	0.92	0.74	0.74	0.75
Subject 2	dys lower	189	KNN	ALL	0.94	0.93	0.93	0.93
	dys upper	186	KNN	SFS + HYP	0.88	0.75	0.73	0.77
Subject 4	dys lower	120	KNN	ALL	1	0.74	0.87	0.64
	dys upper	125	KNN	SFS	0.97	0.70	0.85	0.60
Subject 5	dys lower	338	KNN	ALL	1	0.66	0.96	0.50
	dys upper	441	KNN	SFS	0.98	0.96	0.95	0.98
Subject 6	dys lower	66	n/a	n/a	n/a	n/a	n/a	n/a
	dys upper	66	KNN	ALL + HYP	0.96	0.60	0.65	0.73
Subject 7	dys lower	334	KNN	ALL	0.95	0.82	0.81	0.83
	dys upper	336	NB	ALL + HYP	0.97	0.59	0.73	0.50
Subject 8	dys lower	336	NB	ALL + HYP	1	0.62	0.81	0.50
	dys upper	298	KNN	SFS	0.93	0.64	0.73	0.58
Subject 9	dys lower	588	KNN	ALL + HYP	0.93	0.85	0.84	0.85
	dys upper	583	KNN	ALL	0.97	0.75	0.86	0.66
Subject 10	dys lower	514	n/a	n/a	n/a	n/a	1	1
	dys upper	510	KNN	SFS	0.97	0.53	0.53	0.54
Subject 11	dys lower	478	KNN	ALL + HYP	0.97	0.37	0.43	0.33
	dys upper	444	ENS	ALL	0.84	0.76	0.75	0.77
Subject 12	dys lower	775	KNN	SFS	0.93	0.51	0.61	0.44
	dys upper	1237	ENS	ALL	0.85	0.46	0.54	0.41

Dys lower = dystonia of lower extremity; Dys upper = Dystonia of upper extremity; NB = Naïve Bayes; KNN = k-nearest neighbors; ENS = Ensemble Learning; ALL = all features; SFS = Sequential feature selection; HYP = hyperparameter tuning.

**Table 5 sensors-22-04386-t005:** Overview of mean F1 score, precision and recall.

Dataset	Mean F1 Score Validation	Mean F1 Score Test	Mean Precision Test	Mean Recall Test
dys lower	0.97 ± 0.03	0.67 ± 0.19	0.82 ± 0.18	0.66 ± 0.26
dys upper	0.93 ± 0.06	0.68± 0.14	0.73 ± 0.13	0.66 ± 0.16

Dys lower = dystonia of lower extremity; Dys upper = Dystonia of upper extremity.

**Table 6 sensors-22-04386-t006:** Overview of best generalized model per dataset.

Dataset	Samples	Best Algorithm	Model	F1 Score Validation	F1 Score Test	Precision Test	Recall Test
dys lower	4533	ENS	SFS	0.93	0.45	0.43	0.48
dys upper	4976	KNN	SFS	0.91	0.34	0.32	0.36

Dys lower = dystonia of lower extremity; Dys upper = Dystonia of upper extremity; KNN = k-nearest neighbors; ENS = Ensemble Learning; SFS = Sequential feature selection.

## Data Availability

The dataset analyzed during the current study and code is available on zenodo. Dataset: https://doi.org/10.5281/zenodo.6379451 (accessed on 3 May 2022); Code: https://doi.org/10.5281/zenodo.6379348 (accessed on 3 May 2022).

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
