# Peer review of "Home-Based Measurements of Dystonia in Cerebral Palsy Using Smartphone-Coupled Inertial Sensor Technology and Machine Learning: A Proof-of-Concept Study"

_sensors, 2022, doi:10.3390/s22124386_

Round 1

Reviewer 1 Report

The authors proposed a method using machine learning and deep learning to detect the severity of dystonia from video data. The research approach is very innovative and interesting. Nevertheless, the following specific questions should be answered and added to the manuscript to improve its clarity.
 There are some comments listed below:

  1. The authors mentioned "The average F1 scores (0.67±0.16) in independent test datasets" in the abstract. However, found no matching data in the manuscript. Please justify with relevant experimental data.
  2. Table 2 has a misleading caption, "Types of models training, validated, and tested." It seems to be the performance metric or metrics for error measuring. However, the manuscript mentions it as a type of model training, etc.
  3. Figure 1 and the abstract mention that 24 machine learning models are trained, but in section 2.7, only six machine learning algorithms are mentioned. Please justify.
    Moreover, the paper uses four feature sets (ALL, ALL+HYP, SFS, SFS+HYP) to train those six selected machine learning algorithms. Figure 1 and other manuscript sections should include the same to make it clear and remove any ambiguities.
  4. In section 2.8, the authors mentioned, "For an objective evaluation of the machine learning algorithms, the datasets were divided into a training dataset, validation dataset, and testing dataset (Fig. 2)." However, Figure 2 is a confusion matrix, and the result for other algorithms should be mentioned. Please justify.
  5. Table 3 has no result for the Deep Learning model. Please modify. Moreover, no Deep learning section was found in the manuscript. Please add it or justify it.
  6. In table 4, what do "dys lower" and "dys upper" mean?
  7. In Table 5, the authors mentioned "High F1 scores (0.95±0.05)" and "F1 scores (0.67±0.16).", but no matching data was found in Table 5. Please modify.
  8. No data is available in supplement 3. Please add it in supporting information.

Reviewer 2 Report

Brief summary: The present study deals with the use of home-based measurements to assess the severity of dystonia in patients with cerebral palsy. The proposed system is based on a smartphone with inertial sensors and machine learning algorithm. Twelve patients have been enrolled for the present study and both video and sensor data were collected during activities and rest. Experiments have been conducted outside the clinical settings, in order to overcome limitations due to the laboratory. Clinical scores and sensor data were used to train different machine learning models. The average F1 scores registered in different independent test datasets indicate that it is possible to detected dystonia automatically using individually trained models.

Broad comments: The present work is generally well described and written. The topic well fits current researches and interest in home monitoring of movement disorders, the use of wearable devices and machine learning algorithms. Additional references and literature background should be added in the introduction in order to give a complete overview of the topic. Some additional information and descriptions must be added in the methodology section. Pictures of subjects wearing the systems and performing the task could be added. In the following section I give specific suggestions referred to the text in order to improve the completeness of the work, to better emphasize explicit points and to simplify the comprehension of some concepts.

Specific comments:

  1. In the keywords, use “gait cycle” instead of “cyclogram” and “principal Component Analysis” instead of “PCA”. Consider the possibility to add “wearable device” as keyword.
  2. Line 46-49: the definition and description of cerebral palsy can be improved with additional details.
  3. Line 55-57: please, give some deeper information about the cited treatments
  4. Line 71-79: Additional information, description and current research on wearable sensors must be added to the test. It is important to stress their advantages non-invasiveness, home monitoring, easy to use, small and compact, …) compared to the gold standard stereophotogrammetry, the current applications in clinics, but also the limitations. Similar considerations can be done for the description of machine learning applications. Here some suggestions of additional references that can be reported:
  • Panero, Elisa, et al. "Effect of Deep Brain Stimulation Frequency on Gait Symmetry, Smoothness and Variability using IMU." 2021 IEEE International Symposium on Medical Measurements and Applications (MeMeA). IEEE, 2021.
  • Khaksar, Siavash, et al. "Application of Inertial Measurement Units and Machine Learning Classification in Cerebral Palsy: Randomized Controlled Trial." JMIR rehabilitation and assistive technologies4 (2021): e29769.
  • Silva, Nelson, et al. "The future of General Movement Assessment: The role of computer vision and machine learning–A scoping review." Research in developmental disabilities110 (2021): 103854.
  1. Line 79-84: despite no previous analysis on algorithm specific for automatic evaluation of dystonia in cerebral palsy have been conducted, could you briefly summed up previous and current research on this type of pathology? It allows the reader to understand current interest and open challenges. The authors can describe and cite recent literature studies to support their point of view. Here some suggestions of additional references that can be reported:
  • Schwartz, Michael H., Andrew J. Ries, and Andrew G. Georgiadis. "Estimating the Efficacy of Common Treatments in Children and Young Adults Diagnosed with Cerebral Palsy Using Three Machine Learning Algorithms." medRxiv(2021).
  • Morbidoni, Christian, et al. "Machine-Learning-Based Prediction of Gait Events From EMG in Cerebral Palsy Children." IEEE Transactions on Neural Systems and Rehabilitation Engineering29 (2021): 819-830.

  1. Line 85-87: In this part the principal aim of the study must be reported. It is the crucial part of the introduction section, allowing the reader to understand the limitations of previous research, the necessity to conduct additional and/or different studies and the main innovative scope of the present research Please, revise these two lines in order to give more importance and deeper description of the principal aim of the study.
  2. Line 85: please, avoid the use of “we” as subject. Try to use impersonal form of the sentence. This correction must be applied on the entire manuscript.
  3. Figure 1: the quality of the picture is low, try to use .svg format. Moreover, the small dimension is not helpful for the reading of the content.
  4. Line 100-104: consider the possibility to use a table or a bullet list for the description of participants.
  5. Line 133-135: in my opinion, an additional picture with a patient wearing the sensors might be helpful.
  6. Line 166: consider the possibility to add a graphical representation of acquired signals.
  7. The study presents numerous limitations, as the authors describe in the discussion session. Nevertheless, the proposed investigation is in line with current researches and can be deeply investigated with additional future studies.

Round 2

Reviewer 1 Report

 I am satisfied with the answers and additional explanations of the authors in the revised version of this manuscript, and I suggest it be accepted. 

Reviewer 2 Report

The authors deeply revised the manuscript based on previous comments and corrections. Considering the English form, some minor corrections can be done before submitting the final version.